META-RESEARCH

# Journal policies and editors' opinions on peer review

**Abstract** Peer review practices differ substantially between journals and disciplines. This study presents the results of a survey of 322 editors of journals in ecology, economics, medicine, physics and psychology. We found that 49% of the journals surveyed checked all manuscripts for plagiarism, that 61% allowed authors to recommend both for and against specific reviewers, and that less than 6% used a form of open peer review. Most journals did not have an official policy on altering reports from reviewers, but 91% of editors identified at least one situation in which it was appropriate for an editor to alter a report. Editors were also asked for their views on five issues related to publication ethics. A majority expressed support for co-reviewing, reviewers requesting access to data, reviewers recommending citations to their work, editors publishing in their own journals, and replication studies. Our results provide a window into what is largely an opaque aspect of the scientific process. We hope the findings will inform the debate about the role and transparency of peer review in scholarly publishing.

**DANIEL G HAMILTON\*, HANNAH FRASER, RINK HOEKSTRA AND FIONA FIDLER**

**\*For correspondence:**
hamilton.d@unimelb.edu.au

**Competing interests:** The authors declare that no competing interests exist.

## Introduction

Almost all scientists who pursue publication of their research via academic journals will be familiar with the scrutiny of their work by their peers. This process of journal-organised, pre-publication evaluation to guide editorial decision-making, commonly referred to as 'peer review', is a standard procedure currently employed by over 65,000 English-language journals (*Ulrichsweb, 2020*). However, beyond this broad definition, it is unclear exactly what peer review entails in practice.

Peer review comes in many forms, with each journal adopting their own unique set of policies governing aspects of the process such as: the amount and range of expertise solicited, the level of anonymity afforded, the availability of documentation to reviewers and the readership and the degree of interaction between stakeholders. Given the role peer review ostensibly plays in quality control, and the diversity of models currently in use, understanding what peer review is, and what it does, is considered a high priority (*Tennant and Ross-Hellauer, 2020*).

Despite its diverse, and customarily closed nature, peer review continues to be widely perceived as a valuable endeavour that adds credibility to scholarship by both researchers (*Nicholas et al., 2015*) and the public (*Pew Research Center, 2019*). It is also widely viewed as a process that improves research quality and weeds out irrelevant and flawed work (*Nicholas et al., 2015*; *Taylor and Francis, 2015*; *Ware and Publishing Research Consortium, 2016*). It is likely for these reasons that most researchers see peer review as part of their job, and consequently volunteer 68.5 million hours a year globally on reviewing (*Publons, 2018*).

Most empirical research on peer review has centred on investigating criticisms of it being 'an ineffective, slow, expensive, biased, inefficient, anti-innovatory, easily-abused lottery' (*Smith, 2010*). However, research efforts have done little to allay concerns, with studies reporting low error detection rates by reviewers, inconsistency among reviewers' recommendations, and biases favouring prestigious institutions and established theories (*Nylenna et al., 1994*; *Schroter et al., 2004*; *Schroter et al., 2008*; *Mahoney, 1977*; *Peters and Ceci, 1982*; *Kravitz et al., 2010*). Scholars also point to

examples of important scientific breakthroughs that were initially rejected following peer review, such as the development of the radioimmunoassay (*Yalow, 1978*), as well as peer-reviewed articles that were later retracted due to significant quality concerns, such as the recent high-profile articles in *The Lancet* and *The New England Journal of Medicine* that reported on the utility of chloroquine treatments for COVID-19 (*Mehra et al., 2020a*; *Mehra et al., 2020b*). We also see instances of authors, reviewers and editors subverting and abusing the process, such as cases of authors rigging peer review (*Ferguson et al., 2014*; *Hopp and Hoover, 2017*), reviewers (and editors) manipulating performance metrics by coercing citations (*Hopp and Hoover, 2017*; *Thombs et al., 2015*; *Fong and Wilhite, 2017*; *Ho et al., 2013*) and editors preferentially publishing their research in the journals they edit (*Luty et al., 2009*; *Mani et al., 2013*; *Shelomi, 2014*). Such examples, while perhaps isolated cases, are rhetorically powerful in creating distrust in the peer review process.

Criticism of the "black box" nature of peer review has also been a topic of interest since at least the 1990s and is associated with a large body of literature discussing the merits of "opening up" various aspects of the process. Some research has focussed on "open peer review": most commonly referencing peer review policies that require full disclosure of the identities of authors and reviewers, increased interactions between all stakeholders and publication of review documentation (*Ross-Hellauer, 2017*), while other research has focussed on improving the reporting of journal policies to the scientific community (*Horbach and Halffman, 2020*; *Klebel et al., 2020*). Three such ongoing initiatives include the Transparency in Scholarly Publishing for Open Scholarship Evolution (TRANSPOSE) project (*Klebel et al., 2020*), the Transparency and Openness Promotion (TOP) Guidelines (*Nosek et al., 2015*) and the European Commission's Open Science Monitor (*Parks and Gunashekar, 2017*).

In the current study, we survey editors of ecology, economics, medicine, physics and psychology journals about how they employ peer review. Amongst other things, we investigated the degree of anonymity and interaction between stakeholders, the availability of review documentation, and the outsourcing of peer review. We also collected information about policies on recommending reviewers, voluntary disclosure of identities, editors altering reviewers' reports, and the sharing of data, materials and code. In addition, we sought editors' views on a number of issues in academic publishing: the appropriateness of reviewers co-writing reviews with colleagues or "co-reviewing" (*McDowell et al., 2019*); reviewers suggesting citations to their work; and editors publishing research in the journals they edit. We also gauged editors' views on reviewers requesting access to raw data during review, the value and role of replication studies, and innovation of their own peer review procedures.

## Study participants

A total of 1490 unique editors representing 1500 journals (eight editors represented more than one journal) were invited to participate in this study, which involved two surveys. The first survey (Survey A) contained questions about peer review policies and practices; the second survey (Survey B) covered five issues related to publication ethics (see Materials and methods for further information). In the 13.5 weeks between invitation and deactivation, 336 unique editors entered Survey A, of which 332 consented to participate. Of the 332 journal editors that consented, 300 completed the survey, 22 started but didn't finish, eight opened the survey but provided no responses and two withdrew consent following completion, giving an overall response rate of 21% (322/1500). Following completion of Survey A, 233 (78%) entered and finished Survey B.

Of the 322 editors who provided at least one response to Survey A, 293 (91%) identified themselves as the incoming or outgoing lead editor. The target response rate of 17% (N = 50) was achieved in the ecology (N = 90), psychology (N = 84) and economics (N = 80) groups, but not in the medicine (N = 40) or physics (N = 28) groups (*Figure 1—figure supplement 1*). The distribution of impact factors among invited and participating journals by discipline is displayed in *Figure 1—figure supplement 2*.

## Results: Peer review policies and practices

### Pre-review policies
Just under half of editors reported that their journal routinely checks *all* incoming manuscripts for plagiarism (49%, 154, *Table 1*). The lowest rate of uniform plagiarism checks was reported in the economics group (31%, 24 of 78). The majority of journal editors reported that their journal allows authors to recommend both for and against specific reviewers (61%, 197), with just under a quarter (23%, 73) not providing any routine avenue for authors to influence who reviews their article. One medical journal and one physics journal also reported outsourcing peer review to a commercial third party.

### Interactions and blinding systems
While most editors reported that their journals encourage interaction between reviewers and the handling editor (73%, 230), few reported encouraging dialogue between fellow reviewers (2%, 7) and authors and reviewers (6%, 20). As to blinding procedures (see *Table 2*), we note the predominance of single-blind systems (author identities are known to the reviewers, but the identities of the reviewers are not known to the authors). We also note that 16 editors (5%) reported using hybrid systems in which

authors have the option to conceal their identity in a single-blind system, or reveal it in a double-blind system where authors' and reviewers' identities are hidden from each other. Similarly, 55 editors (18%) of blinded journals reported that reviewers are free to reveal their identities to authors if they wish.

Lastly, 24 (8%) and 19 (6%) editors reported that their journal accepts Registered Reports or uses results-free review respectively i.e. uses 'results-blind' review (*Button et al., 2016*). Registered Reports were most commonly offered by psychology journals (20%, 16 of 82) and results-free review most often used by economics journals (11%, 8 of 72).

### Peer review documentation
Only four editors (1%) reported publishing reviewer reports (signed or unsigned) and decision letters. However, while *public* sharing of peer review documentation remains rare, most editors did report that both reviewer reports (79%, 199) and editorial decision letters (82%, 233) were shared with all reviewers.

Editors were also asked whether an editor at their journal would 'be permitted to edit a reviewer's report' under certain circumstances, and the process they would follow in that case. While most editors stated that their journal didn't have an official policy on editing reports

**Table 1.** Pre-review policies for all journals and by discipline.

| | All journals | | Ecology | | Psychology | | Economics | | Medicine | | Physics | |
|---|---|---|---|---|---|---|---|---|---|---|---|---|
| | N | % | N | % | N | % | N | % | N | % | N | % |
| Plagiarism software usage (N=317) | | | | | | | | | | | | |
| Never | 7 | 2 | 4 | 4 | 1 | 1 | 1 | 1 | 0 | 0 | 1 | 4 |
| Always | 154 | 49 | 46 | 51 | 45 | 54 | 24 | 31 | 25 | 64 | 14 | 54 |
| If suspicion has been raised | 84 | 26 | 24 | 27 | 19 | 23 | 30 | 38 | 4 | 10 | 7 | 27 |
| At editor's discretion | 54 | 17 | 14 | 16 | 13 | 15 | 18 | 23 | 5 | 13 | 4 | 15 |
| I don't know | 5 | 2 | 0 | 0 | 3 | 4 | 2 | 3 | 0 | 0 | 0 | 0 |
| Other | 13 | 4 | 2 | 2 | 3 | 4 | 3 | 4 | 5 | 13 | 0 | 0 |
| Recommending reviewers (N=321) | | | | | | | | | | | | |
| No | 73 | 23 | 0 | 0 | 18 | 21 | 47 | 59 | 4 | 10 | 4 | 14 |
| Yes - Recommend for only | 27 | 8 | 11 | 12 | 6 | 7 | 2 | 3 | 4 | 10 | 4 | 14 |
| Yes - Recommend against only | 12 | 4 | 3 | 3 | 5 | 6 | 3 | 4 | 1 | 2 | 0 | 0 |
| Yes - Recommend for and against | 197 | 61 | 75 | 83 | 51 | 61 | 23 | 29 | 29 | 72 | 19 | 68 |
| Other | 12 | 4 | 1 | 1 | 4 | 5 | 4 | 5 | 2 | 5 | 1 | 4 |
| Outsourcing peer review (N=318) | | | | | | | | | | | | |
| No | 315 | 99 | 90 | 100 | 82 | 100 | 78 | 100 | 38 | 95 | 27 | 96 |
| Yes | 2 | 1 | 0 | 0 | 0 | 0 | 0 | 0 | 1 | 2 | 1 | 4 |
| Other | 1 | 0 | 0 | 0 | 0 | 0 | 0 | 0 | 1 | 2 | 0 | 0 |

**Table 2.** Blinding policies for all journals and by discipline.

| | All journals | | Ecology | | Psychology | | Economics | | Medicine | | Physics | |
|---|---|---|---|---|---|---|---|---|---|---|---|---|
| | N | % | N | % | N | % | N | % | N | % | N | % |
| Open identities | 3 | 1 | 0 | 0 | 0 | 0 | 1 | 1 | 2 | 5 | 0 | 0 |
| Single-blind | 176 | 57 | 68 | 78 | 16 | 20 | 33 | 43 | 33 | 87 | 26 | 100 |
| Single-blind (hybrid) | 12 | 4 | 3 | 3 | 8 | 10 | 1 | 1 | 0 | 0 | 0 | 0 |
| Double-blind (hybrid) | 4 | 1 | 1 | 1 | 3 | 4 | 0 | 0 | 0 | 0 | 0 | 0 |
| Double-blind | 109 | 36 | 15 | 17 | 51 | 65 | 40 | 52 | 3 | 8 | 0 | 0 |
| Triple-blind | 3 | 1 | 0 | 0 | 1 | 1 | 2 | 3 | 0 | 0 | 0 | 0 |

(84%, 258), 276 (91%) identified at least one situation where editors at their journal would be permitted to alter a reviewer's report with or without the reviewer's permission (for an overview of possible situations, see *Table 3*). The two most common circumstances where editing was deemed acceptable were when the review contained offensive language (85%, 247) or discriminatory comments (83%, 242). For example, among medical journal editors, 100% of respondents reported it would be acceptable to edit the report under both of these circumstances. Further to this, 39% of responding editors reported that it would be acceptable to edit a review without the reviewer's permission if they identified themselves (e.g. signed their review) in a blinded review system. Beyond removing offensive material, inappropriate references and identifying features, 55 editors (19%) reported that it would be acceptable to edit a reviewer's report if they disagreed with the recommendation; 22 of whom reported this would be acceptable to do so without the reviewer's permission.

*Research output sharing*

The last set of policies in our survey focussed on the sharing of data, research materials and analysis scripts. When asked 'what is the journal's current policy on the availability of research data, materials and code following publication', the two most commonly observed policies were 'Encourages sharing, but it is not required' and 'No policy' (*Table 4*). Editors of 20%, 15% and 14% of surveyed ecology, medical and economics journals, respectively, declared having mandatory data sharing policies. In contrast, mandatory policies were noted at only one of the surveyed psychology journals, and none of the physics journals. While relatively few editors reported policies requiring data to be shared, 52 (18%) reported that they mandate including a data availability statement specifying whether any data will be shared, and if so, how to access it.

## Results: Editors' views on publication ethics

Participating editors were also asked about their views on five issues related to publication ethics

**Table 3.** Situations where an editor may edit a reviewer's report.

| | Never acceptable to edit the report | | Acceptable to edit without reviewer's permission | | Acceptable to edit, but only with reviewer's permission | |
|---|---|---|---|---|---|---|
| | N | % | N | % | N | % |
| When a reviewer identifies themselves in a blinded peer review framework (N=276) | 93 | 34 | 109 | 39 | 74 | 27 |
| When a reviewer has used inappropriate or offensive language (N=291) | 44 | 15 | 170 | 58 | 77 | 26 |
| When the reviewer has made an inappropriate reference to an author's gender, age etc (N=290) | 48 | 17 | 163 | 56 | 79 | 27 |
| When there are spelling and/or grammatical errors in the review (N=294) | 104 | 35 | 141 | 48 | 49 | 17 |
| When the review has English language problems (N=292) | 95 | 33 | 124 | 42 | 73 | 25 |
| When the reviewer has left in their comments to the editor (N-290) | 50 | 17 | 179 | 62 | 61 | 21 |
| When the editor disagrees with the reviewer's recommendation (N=293) | 238 | 81 | 22 | 8 | 33 | 11 |

**Table 4.** Journal policies on the sharing of research data, materials and code.

| | Research data (N=294) | | Research materials (N=264) | | Research code (N=255) | |
|---|---|---|---|---|---|---|
| | N | % | N | % | N | % |
| Encourages sharing but it is not required | 168 | 57 | 143 | 54 | 133 | 52 |
| Must make available if requested | 41 | 14 | 29 | 11 | 32 | 13 |
| In-text statement required | 52 | 18 | 34 | 13 | 29 | 11 |
| Requires posting to a trusted repository | 34 | 12 | 16 | 6 | 19 | 7 |
| No policy | 65 | 22 | 69 | 26 | 65 | 25 |
| Not applicable | 10 | 3 | 13 | 5 | 11 | 4 |
| I don't know | 4 | 1 | 4 | 2 | 6 | 2 |
| Other | 8 | 3 | 7 | 3 | 7 | 3 |

Percentages do not add up to 100% due to multiple answers being possible

issues, and asked to describe what, if anything, they would change about how their journal conducts peer review (*Figure 1*).

### Co-writing reviews with colleagues (co-reviewing)

Of the 203 editors that provided a codable position on the topic, 88 (43%) personally encouraged co-reviewing, 70 (34%) reported that it is an acceptable practice at their journal, and 4 (2%) responded that they would allow it despite personally discouraging the practice. Of the editors who expressed support for co-reviewing, one in three added that they think it is a good way to develop the next generation of reviewers. The belief that co-reviewing results in a better review and helps connect journals with future reviewers and editors were also cited as reasons to support the practice. In contrast, editors who discouraged the practice, expressed disinterest in the written opinions of reviewers they did not solicit themselves, as well as concerns about decreased review quality and efficiency.

While endorsement of co-reviewing was high, editors often qualified the need to disclose the practice to ensure co-reviewers are credited for their work, as well as to assess potential conflicts. Oversight over the process by the editor-invited reviewer, as well as ensuring adherence to confidentiality and conflicts of interest policies were also strongly emphasised. Concerns around breaches of these same themes were cited by editors as reasons to discourage the practice. The following statements capture common sentiments and caveats concisely:

*"We do it, and I think it is important training. We emphasize that the ultimate responsibility is with the mentor, that all COI policies apply to the mentor and the student, and that confidentiality extends no further than one student with the invited reviewer."* (Ecology journal editor).

*"I believe this is an incredibly valuable process, providing that it is true "co-writing" and not just getting people to do your work and taking credit for it. This is a great way to train learners in how to properly review, something that is not done enough."* (Medical journal editor).

*". . . I feel this would violate confidentiality and open submitters to the risk of leaking their manuscript or parts of their submission that the authors might find problematic. Also, we ask for the reviewer's expertise purposefully. This is not intended to be a pedagogic exercise."* (Psychology journal editor).

### Reviewers suggesting citations of their work

Fewer than 5% of editors overall objected to reviewers suggesting citations of their own work. In fact, a common view was that this should be expected given invited reviewers have likely done relevant work. Despite few objections to this practice, respondents expressed the need for editors to stay vigilant and arbitrate cases. For example, editors were often described as responsible for ensuring that recommended citations are relevant, address significant gaps or mischaracterisations, and display balance or restraint (often to prevent unblinding of reviewers). A common view was that editors should ultimately leave it to authors to determine whether to cite the suggested paper or

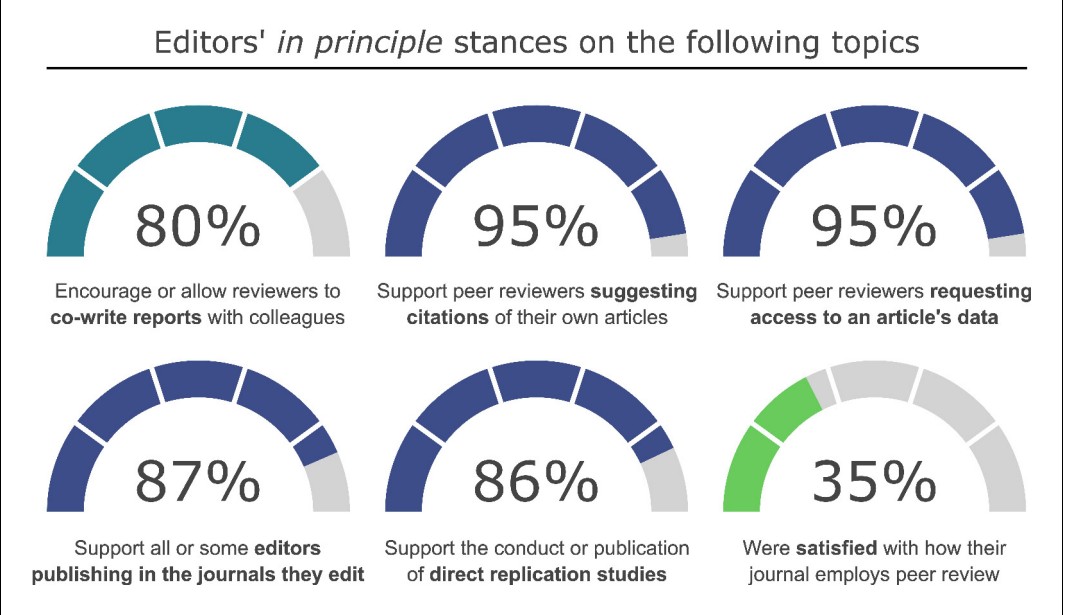

**Figure 1.** Participating editors' *in principle* stances on the six topics raised in Survey B. The figures presented are limited to statements that provided a clear view for or against the topic of interest. An interactive version of this figure reporting results by discipline can be viewed at https://plotly.com/~dghamilton/9/ (*Supplementary file 1*). Source data for the figure can be found at https://doi.org/10.17605/osf.io/cy2re.
The online version of this article includes the following figure supplement(s) for figure 1:

**Figure supplement 1.** Survey response rate by discipline.
**Figure supplement 2.** Distribution of impact factors among invited and participating journals by discipline.

not. Interestingly, three editors also stated that gratuitous requests would be grounds for editing a report. The following statements touch upon many of these themes:

"... *We invite people to review because they are subject matter experts, and because they are experts, they have often done relevant work. However, when a reviewer does nothing but add citations of their own work, that is clearly inappropriate. We often state in the decision letter that it is not necessary to cite all requested references ... or, because all our reviewers are aware that we may edit their comments, we will take requests for inappropriate citations out of decision letters when excessive. ..."* (Medical journal editor).

"*This is borderline, and acceptable only when this is relevant to the paper. This is a case for editing the review*" (Physics journal editor).

"*Yes. But only if their work is so significant on the topic that to leave it out would be a significant indication of a large scholarly oversight. I do dislike this practice and often discontinue the use of the reviewer if the suggestions are not germane or if it happens consistently.*" (Economics journal editor).

### Reviewers requesting access to raw data

Of the 210 editors that provided codable views, 200 (95%) stated that they would support and mediate a request from a reviewer to access a manuscript's raw data if they felt it was needed. This high level of support was noted to be consistent across all disciplines. However, of these respondents, approximately one in six added that they would need a compelling reason to do so, such as if credible ethical or quality concerns were raised. Furthermore, despite the high levels of *in principle* support, some also conceded that respecting data protections would need to be prioritised. Some editors expressed that while they would attempt to circumnavigate confidentiality issues by offering investigation through an independent intermediary, disclosure would ultimately be left at the authors' discretion. Consequently, few editors outside those of journals adopting mandatory data sharing policies stated that manuscript acceptance would be contingent on sharing. Some of these views were captured in the following responses:

"*Absolutely. But I've also indicated that this is at the authors' discretion and have also conveyed any problems with Human Ethics in*

sharing a data set when that has been an issue. Both parties must agree to confidentiality to avoid leaking if the author does not or cannot provide a full data set for public view." (Psychology journal editor).

"I would definitely support and mediate the request of reviewer to see raw data. The goal of an editor is to publish solid and reproducible science and we have to help reviewers provide a fair and unbiased review based on the scientific results." (Medical journal editor).

"I definitely would support a request from a peer reviewer to see a manuscript's raw data if they feel it's necessary. Acquisition, processing and (statistical) evaluation of data in ecology is often complex, and the reviewer can assess the validity of data interpretation sometimes only when seeing original data and experimental design." (Ecology journal editor).

### Editors publishing in the journals they edit

When asked how often an editor should publish their own original articles in a journal they edit, 176 (79%) editors described a scenario where they thought it would be acceptable for any editor to do so. A smaller proportion of respondents stated that this would not be acceptable for the lead editor (8%, 17), or for any editor (13%, 29). Once again, support was often qualified with prerequisites, most commonly processes to protect against conflicts of interest. Suggested processes included: independent editing and reviewing, preventing the submitting editor knowing the identities of the handling editor and reviewers (and vice-versa), and providing a statement declaring the handling editor's identity on the final article.

Interestingly, views on how often an editor should submit research were more polarised among respondents, with one group expressing that it should not be restricted, and another suggesting that manuscripts should only be submitted rarely. However, despite these opposing views on frequency, there was a strong sentiment around the lack of fairness associated with preventing editors publishing in their journals among both groups. Many respondents discussed the negative consequences, such as: shutting out potentially desirable content, restricting publication venues for collaborators and deterring researchers taking on editorial roles. In contrast, many editors stated that publishing in one's journals demonstrates a commitment to it. These beliefs were captured well in the following statements:

"As often as the work meets the standard. ... Telling people that they cannot publish in it lowers the value of what is a thankless task for which we aren't adequately compensated anyway." (Economics journal editor).

"Whenever the subject matter is appropriate. It is crazy to shut off an excellent journal from an excellent researcher when they put a lot of their own time and effort into ensuring the journal succeeds. It is easy to set up a process whereby that editor never sees their own work, or reviews, or reviewer identity - or even handling editor identity, so that the process can be independent and not influenced by the author/editor." (Ecology journal editor).

"Not often. ... I have been reluctant to send my work to the journal even if a former editor were to decision it. However, I may consider sending a paper or two in my [next] term. I have published many papers in my journal prior to becoming editor so I have proven that I can do it. Thus, perhaps it is not fair on myself to limit where I can publish. It can also be damaging to the journal to the extent that the work is very much suited for the [readership]. But in all cases, it must be done within reason and very carefully ..." (Psychology journal editor).

"Not very often. In [my country], though, there are not so many (recognized) outlets for economists, while there is a need for decent local work to be done. Thus, I do publish an article here and there. I have absolutely no access to the article while it is under review; so, at least in principle, I cannot influence its 'passage' through the system." (Economics journal editor).

### Direct replications

When asked their views on direct replication studies (i.e. research that follows the methods of another study as closely as possible), of the 183 editors that provided a clear stance, 86% stated that they support the practice or allow replication studies to be submitted to their journals. However, while respondents often perceived direct replications as an important and informative line of research, many also stressed the need for replication studies to be accompanied by novel work or offer new insights as a condition for publication, with the discovery of findings contradicting or undermining the original study being preferable for some. Similarly, the importance of the original study, as well as the number of existing replication attempts were also highlighted as important factors influencing the likelihood of acceptance of direct replication studies.

In contrast, the few editors who discouraged, or were ambivalent towards replication studies tended not to explain why they held these beliefs. When reasons were given, they most often related to perceived lack of interest from the readership, concerns surrounding potential agendas by researchers, and non-compliance with their journal's scope for novel material. Consequently, many editors encouraged submission of research deliberately altering elements of the original study to test hypotheses in a new way, or to test generalisability of results in other settings i.e. 'conceptual' replication. The range of difficult considerations contributing to editors' perceptions of replication studies are illustrated in the following statements:

"I am very conflicted about this. It is important to get to the truth, but there are so many reasons an experiment might not work, and many of them have nothing to do with truth. And, frankly, as an editor, I am completely uninterested in publishing replication studies. No one will read the journal. So this is a very tough issue. I guess I'm more in favor of people attacking the same question from different angles to get at the truth, rather than attempting to perform precise replication. It is also really critical that those performing the replication studies do so ingenuously, and not with an agenda to disprove the original, or any intellectual conflict of interest. I find this is rarely the case - agendas are rife in replication work. ..." (Medical journal editor).

"Replication studies using new data are extremely valuable. Whether it would be published in my journal would depend on the perceived value of the study being replicated, the scope and quality of the analysis and whether the replication supports the existing finding. ..." (Economics journal editor).

"A direct, close replication is interesting and worthwhile. A journal has no particular duty to publish these – I reject the 'Pottery Barn Rule'. But if they're interesting, informative and well-done, why wouldn't a journal want to publish them?" (Psychology journal editor).

"Replication studies are essential as one of the points of scientific research is to be able replicate the results of other research. An inability to replicate a study should make us question the results of that study. Journals like [redacted] have a section devoted to Replication Studies with an Editor in charge of these studies. This is best practice as far as I am concerned." (Economics journal editor).

### Changes to existing peer review practices

Lastly, when asked if they would change anything about how their journal conducts peer review, of the 196 editors who responded, only 35% indicated that they were satisfied with the current system. The remaining two-thirds discussed changes they were contemplating, implementing or had implemented recently. The most frequently mentioned changes included: modifying the blinding system (most often increasing anonymity); improving how the journal finds reviewers; and improving the overall efficiency of the process. Another common theme was incentives and rewards for reviewers: the ideas put forward included remuneration, acknowledgment via Publons, the waiving of article processing charges, and professional development credits.

## Discussion

Innovation adoption is a complex and active process that regularly begins with the recognition of a need, or opportunities to improve processes (*Horbach and Halffman, 2020*; *Wisdom et al., 2014*). When considering innovation of peer review, the need to maintain, and the desire to expand and incentivise their journal's pool of reviewers featured prominently among editors in the current study. These motivations were often explicitly cited as reasons to justify policy-making decisions and positions on the raised ethical issues.

At times, these needs and opportunities aligned. For example, co-reviewing was commonly endorsed because it presented as an opportunity to connect with and train the next generation of reviewers. However, at other times they were noted to be in conflict, stifling potential change. For example, of the editors who supported reviewers requesting access to data, many were also conflicted about making acceptance contingent upon sharing. Reluctance among editors to dictate such terms to authors may also help explain the low uptake of mandatory data sharing policies observed in both the current and previous research (*Resnik et al., 2019*). However, setting aside issues of low adherence to journal and funder instructions (*Alsheikh-Ali et al., 2011*; *Rowhani-Farid and Barnett, 2016*; *Couture et al., 2018*), it is likely that uptake of mandatory data sharing policies will increase in the future as major funding agencies like the National Institutes of Health (NIH) and the Medical Research Council continue to implement increasingly strict policies on data

sharing (*National Institutes of Health (NIH), 2020*; *Medical Research Council, 2016*).

A further example included the conflict between an editor's desire to increase transparency and the fear of alienating reviewers. With regards to increasing transparency through the adoption of open peer review policies, there is evidence to substantiate such fears (*Publons, 2018*; *Ho et al., 2013*; *van Rooyen et al., 1999*; *van Rooyen et al., 2010*; *Ware, 2008*; *Walsh et al., 2000*). For example, findings from a survey of over 12,000 researchers by Publons in 2018 reported that 37% to 49% of participants would be less likely to accept an invitation to review if open policies were adopted (*Publons, 2018*). However, there is also evidence to suggest these views may be shifting, especially among younger and non-academic scholars (*Publons, 2018*; *Bravo et al., 2019*).

Previous research has also reported that three quarters of editors consider finding reviewers the most difficult part of their job, a task which is projected to become more difficult in the future (*Publons, 2018*). Given this, in addition to the estimate that 10% of reviewers account for 50% of performed reviews (*Publons, 2018*), it is likely that the perceived impact of policy changes on an editor's ability to recruit reviewers contributes heavily to journal policy-making decisions. These factors may further help to explain the low implementation of the three policies most often associated with open peer review: open identities, open reports and open interactions in the current study and previous research (*Horbach and Halffman, 2020*; *Klebel et al., 2020*; *Parks and Gunashekar, 2017*).

Despite the low popularity of unblinded models, previous research has noted increasing adoption of policies that allow authors and reviewers greater flexibility in deciding whether to disclose their identity or not during peer review (*Klebel et al., 2020*). For example, in our sample, 18% of editors reported that reviewer anonymity was left at the reviewers' discretion. This estimate is consistent with findings from *Klebel et al., 2020*, and almost four-times higher than the highest estimate from Publons data in 2017 (*Parks and Gunashekar, 2017*). However, importantly we note this movement toward greater flexibility appears to be at odds with the 39% of other editors who reported that reviewers identifying themselves within a blinded framework would be grounds for editing the report without the reviewer's permission.

The notion of editors altering reviews, as well as the belief that reviewers are aware of this practice, represented some of the most intriguing findings of the current survey. The high percentage of editors that identified at least one situation where they considered editing a review acceptable (91%) is also consistent with a survey of 146 editors and publishers recently released by the Committee on Publication Ethics (COPE) (86%) (*COPE Council, 2020*). Many of the reasons the COPE survey participants provided also echo those explored in the current survey, including the observation that some respondents viewed removing a reviewer's recommendation as a valid reason to edit a report. Unfortunately, we did not ask editors what edits they envisioned making in each of the proposed situations, so we are hesitant to draw any strong conclusions. For the case of an editor altering a reviewer's report when they disagree with the recommendation, it is plausible that this might mean removing or changing a reviewer's recommendation to accept or reject the paper. It is also plausible that editors may have envisaged more innocuous changes to temper the reviewer's critique, or more extensive changes to the content of the reviewer's commentary. Two members of the authorship team (FF, RH) have experienced an example of the latter situation, which was covered in a recent news article in *Science* (*O'Grady, 2020*).

When considering the notion of editors altering reviews, one could argue that fixing spelling and grammatical errors is benign, or even a necessary copy-editing step prior to publication of reviews. Editors' preference to tone down potentially offensive or discriminatory remarks is also comprehendible, particularly given current pressures to retain reviewers, the non-trivial rate of unprofessional reviews and their potential negative impact on authors' productivity and wellbeing (*Publons, 2018*; *Ho et al., 2013*; *Gerwing et al., 2020*; *Silbiger and Stubler, 2019*). This course of action has also been specifically advised in cases discussed by the COPE Forum (*Committee on Publication Ethics, 2011*). Removal of identifying information is also likely reflective of stricter enforcement of blinding policies (*COPE Council, 2020*). However, what is concerning is the number of editors (8–62% depending on the situation encountered, see *Table 3*) in the current survey that reported it would be acceptable to make these changes without consulting reviewers.

It is also particularly concerning, given such practices would be undetectable in journals that

do not publish reviews or share complete decision letters with reviewers. Of note, 15% of the editors surveyed in the current study and 50% of editors surveyed by Horbach and Halffman reported that their journals do not share decision letters with reviewers (*Horbach and Halffman, 2020*). Ultimately, given the perception that editors and editorial boards are largely responsible for managing ethical issues (*Taylor and Francis, 2015*), such practices, particularly if shown to change reviewer recommendations may act to further degrade trust in peer review and increase calls for greater transparency.

The current study explored a series of peer review policies across a range of scientific disciplines. The survey achieved a high overall response rate (21%) compared to past survey efforts (4–6%) (*Hopp and Hoover, 2017*; *Horbach and Halffman, 2020*), as well as similar distributions of invited and responding journals by impact factor. However, we note some weaknesses of the study. Despite similar distributions by impact factor, we cannot rule out systematic differences between responders and non-responders that may have resulted in an unrepresentative sample. We also did not explore how uptake and reporting of policies differ depending on the journal scope or language (e.g. dedicated review journals, foreign language journals), the journal publishing model (e.g. open access, hybrid or subscription model), or whether the journal is led by an academic or professional editor. Furthermore, the authors note that our study only provides superficial information on selected policies and practices. For example, investigation into *how* editors use author-recommended reviewers, as well as nuances in policies, such as mixed policies that mandate data sharing for some study designs (e.g. clinical trials) or research outputs (e.g. x-ray crystallography), but not others, were outside of the scope of the study. Lastly, the survey represents a snapshot of journal policies across a three-month period in 2019. Consequently, recognising that journal policies change with time and new leadership, the policy landscape could shift substantially through time.

## Conclusion

Due to its closed nature, peer review continues to be difficult to examine. Our study sheds light on what peer review policies and practices are currently being used by journals in ecology, economics, medicine, physics and psychology, and

gauges editor attitudes on some topical publication ethics issues. It highlights the tension between maintaining and growing the reviewer pool, an integral part of an editor's role, and engaging in more open practices, such as open peer review. It also draws attention to the ethics of editors altering reviewers' reports. We hope that the data presented in this study will stimulate further discussion about the role of peer review in scholarly publishing, catalyse future research and contribute to guideline development efforts and policymaking, particularly those concerning transparency in editorial and publishing policy.

## Materials and methods

### Journal selection strategy

Clarivate Analytics' InCites Journal Citation Reports ("Browse by Journal" function) was used to identify journals within five broad fields of interest to members of the authorship team, specifically: ecology, economics, medicine, physics and psychology. Lists of journals sorted by 2017 Journal Impact Factor were generated for each discipline on April 9th, 2019. Journals flagged as duplicate entries across disciplinary lists were subjectively removed from the less subject-appropriate list (e.g. *The Journal of Comparative Psychology* was removed from the ecology list) and then replaced with the next entry. The list of categories used in InCites to define disciplines can be found in *Supplementary file 2*. Foreign-language journals, book series and data repositories were excluded from this study.

Aiming for a minimum of 50 responses per discipline, the top 300 journals in each field (when ranked by impact factor) were invited to participate (assuming an expected response rate of between 15–20% [*Byrne, 2000*]). Therefore, our target response rate was 17%. Impact factors were used to select journals for sampling as they can be reflective of the relative level of visibility of journals to researchers within a field (*Paulus et al., 2018*).

The lead editor for each of the 1500 journals was identified via the journal's website, and contact emails obtained via a reliable source e.g. journals' and academic institutions' websites or recent publications. In the event that the same individual was found to edit multiple journals (duplicate editor) a co-lead editor or deputy editor was invited if available, or the editor was contacted by email and asked whether they

would complete the survey on behalf of each of the journals they edit. As indicated above, in the final sample eight editors represented more than one journal.

### Survey themes and questions

Two separate surveys were created for this study. The first (Survey A) aimed to characterise journals' peer review policies and practices as understood and reported by a member of the journal's editorial team. The second (Survey B) was designed to capture editors' opinions on five current peer review and publication ethics issues. For Survey A, participants were informed prior to commencing that information on routine peer review policies and practices were going to be collected, and provided consent allowing the study team to link responses back to the journal. For Survey B, participants were informed that all statements would be kept anonymous. Participants provided consent prior to beginning each survey. All questions for both surveys can be found in *Supplementary file 3*. De-identified responses to Survey A and coding data for Survey B are publicly available (*Hamilton et al., 2020*). The survey and data collection strategy was approved by the University of Melbourne's Faculty of Science Human Ethics Advisory Group (Project ID: 1954490.1) prior to commencement.

### Survey platform and distribution

Qualtrics Solutions' Online Survey Software (Qualtrics, Provo, UT) was used to create and distribute the two surveys. Email invitations housing a personalised link to Survey A were generated and sent to all editors between June 26th and July 18th, 2019. Participants who completed Survey A were automatically redirected to the anonymous Survey B. Throughout the course of the project, three reminder emails were sent in July, August and September 2019 to editors who had not started (and not opted-out of future email reminders), or not completed Survey A. Both surveys were deactivated, and all responses recorded on October 1 st, 2019.

### Statistical analysis and reporting

Descriptive statistics were used to analyse answers to multiple choice survey questions in R (R Foundation for Statistical Computing, Vienna, Austria, v3.6.0). Missing responses were omitted from the proportions reported in the results section. The full breakdown of Survey A responses by discipline can be seen in *Supplementary file 4*. All figures were created in R using the ggplot2 (v3.2.0), plotly (v4.9.2) and viridis packages (v0.5.1) (*Wickham, 2016*; *Sievert, 2018*; *Garnier, 2018*).

Responses to the open-ended questions in Survey B were analysed in NVivo (QSR International Pty Ltd., v12.4.0). An inductive (data-driven) approach was used to develop analytic categories (codebooks) for qualitative analysis. Preliminary codebooks for each question in Survey B were drafted by one author (DGH) after carefully reading and re-reading all responses to identify emergent themes. This process was repeated until all emergent themes relevant to the research question were captured and analytic categories could not be further refined. All codebooks were then pilot tested on six randomly selected responses by two coders (DGH; HF) for calibration purposes. Following further discussion and refinement, codebooks were then repeatedly tested on 35 (15%) randomly selected responses by the same two coders (DGH; HF) until sufficient inter-coder reliability was obtained.

For the purposes of this study, a codebook was considered stable once an overall unweighted kappa coefficient greater than 0.70, and average percentage agreement greater than 95% was achieved. Codebooks that did not satisfy these criteria after a round of testing were discussed and refined prior to being re-tested. The summarised results of the reliability analysis can be found in *Supplementary file 2*. Once determined to be stable, codebooks were then used by one coder (DGH) to structure the analysis of the full dataset of responses. Results from this analysis were reported in a mixed methods approach by survey question. This included quantitative methods (e.g. counts of editors coded as for or against the issue). In addition, qualitative methods were used such as: exploring the relationships between expressed themes (e.g. identifying reasons for supporting or discouraging a practice) and reporting of contrasting views and deviant cases.

### Acknowledgements

DGH is a PhD candidate supported by an Australian Commonwealth Government Research Training Program Scholarship. The authors thank Dr Eden Smith for advice on the qualitative analysis and feedback on the manuscript. The authors also thank Aurora Marquette, Oscar Marquette and Acacia Finch Buchanan for assistance with data extraction.

**Daniel G Hamilton** is in the Interdisciplinary Metaresearch Group, School of BioSciences, University of Melbourne, Victoria, Australia

hamilton.d@unimelb.edu.au

https://orcid.org/0000-0001-8104-474X

**Hannah Fraser** is in the Interdisciplinary Metaresearch Group, School of BioSciences, University of Melbourne, Victoria, Australia

https://orcid.org/0000-0003-2443-4463

**Rink Hoekstra** is in the Department of Educational Sciences, University of Groningen, Groningen, Netherlands

https://orcid.org/0000-0002-1588-7527

**Fiona Fidler** is in the Interdisciplinary Metaresearch Group, School of BioSciences, and the School of Historical and Philosophical Studies, University of Melbourne, Victoria, Australia

*Author contributions:* Daniel G Hamilton, Conceptualization, Data curation, Formal analysis, Investigation, Visualization, Methodology, Writing - original draft, Project administration; Hannah Fraser, Conceptualization, Formal analysis, Supervision, Methodology, Writing - review and editing; Rink Hoekstra, Fiona Fidler, Conceptualization, Supervision, Methodology, Writing - review and editing

*Competing interests:* The authors declare that no competing interests exist.

*Ethics:* Human subjects: Both surveys were reviewed and approved by the University of Melbourne's Faculty of Science Human Ethics Advisory Group (Project ID: 1954490.1) prior to distribution. All participants were provided detailed information about the purpose of the study, specifically that information on routine peer review policies and practices at their journal, as well as their views on some publication ethics issues were going to be collected. Informed consent was obtained from participants prior to beginning each survey.

## Funding

No external funding was received for this work.

### Decision letter and Author response

Decision letter https://doi.org/10.7554/eLife.62529.sa1
Author response https://doi.org/10.7554/eLife.62529.sa2

## Additional files

### Supplementary files

- Supplementary file 1. Interactive version of *Figure 1*.

- Supplementary file 2. Categories used to define disciplines in InCites; codebook reliability testing.

- Supplementary file 3. Survey questions (A and B).

- Supplementary file 4. Responses to questions in survey A.

- Transparent reporting form

## Data availability

De-identified responses to Survey A and coding data for Survey B are publicly available (https://doi.org/10.17605/OSF.IO/CY2RE).

The following dataset was generated:

| Author(s) | Year | Dataset URL | Database and Identifier |
|---|---|---|---|
| Hamilton DG, Fraser H, Hoekstra R, Fidler F | 2020 | https://doi.org/10.17605/OSF.IO/CY2RE | Open Science Framework, 10.17605/osf.io/cy2re |

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
