## [Decision Letter]

Thank you for submitting your article "Meta-Research: Journal policies and editors' opinions on peer review" to *eLife* for consideration as a Feature Article. Your article has been reviewed by three peer reviewers, including the *eLife* Features Editor Peter Rodgers. The reviewers have discussed the reviews and have drafted this decision letter to help you prepare a revised submission.

The following individuals involved in review of your submission have agreed to reveal their identity: Liz Allen (Reviewer #1); Tony Ross-Hellauer (Reviewer #2).

Summary:

Reviewer #1: This is an interesting descriptive analysis of journal editors' views – albeit, as the authors acknowledge, a snapshot at a time when there are many changes afoot across scholarly publishing. The discussion of how Editors are thinking about the policies and practice around, for example open data, publishing replication studies, and open peer review, are timely and relevant topics with implications for meta-research more generally. There is also a dearth of evidence around aspects of the publishing process and particularly the “black box” that is peer review. However, there are also a number of points that need to be addressed to make the article suitable for publication.

Essential revisions:

1) In terms of the sampling, I wasn't sure why the particular mix of journal subject areas was picked (was this important? why pick these?). It is also not clear why the focus on “high impact journals” is important as part of the sampling frame; the authors do not explain why this was the selection strategy – is it perhaps based upon an assumption that these would be more established journals therefore would have established policies/substantial experience? Or another reason?

2) Did you distinguish between open-access (OA) journals and non-OA journals? If yes, please discuss what you found. If no, please mention this in the section on the limitations of the survey as the issue of OA vs non-OA journals may influence the views of editors on some of the topics covered in the survey.

3) Did you distinguish between journals edited by academic editors and those edited by professional editors? If yes, please discuss what you found. If no, please mention this in the section on the limitations of the survey. The issue of academic vs professional editors may influence some of your findings because, for example, one might expect professional editors to be more familiar with journal policies on data etc, and also to have more time to implement/police these policies.

4) Section on "Reviewers requesting access to raw data".

Do you know how many of the journals in your sample mandate data sharing?

Also, there have been a lot of articles about researchers not being able to get data from published authors, even for papers published in journals that mandate data sharing, and it would be good to cite and discuss some of these articles in the Discussion, and also to mention that more and more funders are insisting on open data. (There are similar problems with journals signing up to guidelines/mandates wrt various aspects of clinical trials, but not enforcing these guidelines/mandates. I know that clinical trials are beyond the scope of this article, but the problems encountered with papers reporting the results of trials highlight the difficulty/complexity of implementing guidelines/mandates).

5) The practice of editors editing reports when they disagree with the recommendation needs to be discussed further – even if it is only to say something along the lines that it is not clear if this involved deleting comments like "I support publication" or "I oppose publication", or if it was more extensive, and that unfortunately there were no open responses on this topic.

Also, please clarify what the figures (8-62%) refer to.

6) The authors could have perhaps included some more explicit recommendations for further analysis and even policy and practice.

7) Figure 1: Please replace Figure 1 with a table A6 from Supplementary file 3.

---

## [Author Response]

Essential revisions:1) In terms of the sampling, I wasn't sure why the particular mix of journal subject areas was picked (was this important? why pick these?).

Thank you very much for the comments. To address the first point, the journal subject areas were chosen primarily as each one relates to a member of the authorship team’s fields of interest (DGH – Medicine & Physics, HF – Ecology, RH – Psychology, FF – Psychology, Ecology & Economics). Furthermore, the authors felt that these fields would suffice for the purposes of the study as they also have the benefit of representing a broad area of scientific disciplines. For example, the five chosen areas cover each of the five main fields of science as used in the CWTS Leiden Ranking, namely: Social Science & Humanities (Psychology, Economics), Biomedical & Health Sciences (Medicine), Physical Sciences & Engineering (Physics), Life & Earth Science (Ecology) and Mathematics & Computer Sciences (Physics). Some text has been added into the Materials and methods section to clarify this to prospective readers.

It is also not clear why the focus on “high impact journals” is important as part of the sampling frame; the authors do not explain why this was the selection strategy – is it perhaps based upon an assumption that these would be more established journals therefore would have established policies/substantial experience? Or another reason?

The decision to limit the focus to the 300 journals with the highest impact factor in each discipline was largely done for reasons of convenience. From a bibliometric point of view, Clarivate Analytics’ “impact factors” are arguably the most commonly used and well-known impact measure in academia. They are also a simple metric to access and interpret. Furthermore, we are the first to acknowledge that we do not believe impact factors are a surrogate for journal quality (or if they are, they’re a poor one). However, we do acknowledge that there is evidence to suggest that they can provide some useful information on the degree of “*visibility*” of journals within a field (Paulus, Cruz and Krach, 2018). Consequently, given that manually assembling and extracting contact information from all journals within a specific field was not feasible, and our overall aim to survey journals that are well-known to researchers within a particular field, we decided that limiting the scope to popular journals indexed in the Web of Science would be a defensible sampling strategy. Some further text has been added to the Materials and methods section to explain this decision (paragraph two).

2) Did you distinguish between open-access (OA) journals and non-OA journals? If yes, please discuss what you found. If no, please mention this in the section on the limitations of the survey as the issue of OA vs non-OA journals may influence the views of editors on some of the topics covered in the survey.

This is another valid point. Unfortunately, we did not collect this information and so list this as a limitation in the Discussion section (paragraph nine).

3) Did you distinguish between journals edited by academic editors and those edited by professional editors? If yes, please discuss what you found. If no, please mention this in the section on the limitations of the survey. The issue of academic vs professional editors may influence some of your findings because, for example, one might expect professional editors to be more familiar with journal policies on data etc, and also to have more time to implement/police these policies.

As above, this is another piece of information we overlooked. A sentence has been added into the limitations to point this out (paragraph nine).

4) Section on "Reviewers requesting access to raw data".Do you know how many of the journals in your sample mandate data sharing?Also, there have been a lot of articles about researchers not being able to get data from published authors, even for papers published in journals that mandate data sharing, and it would be good to cite and discuss some of these articles in the Discussion, and also to mention that more and more funders are insisting on open data. (There are similar problems with journals signing up to guidelines/mandates with regards to various aspects of clinical trials, but not enforcing these guidelines/mandates. I know that clinical trials are beyond the scope of this article, but the problems encountered with papers reporting the results of trials highlight the difficulty/complexity of implementing guidelines/mandates).

Information on how many of the participating journals mandate deposition of data into an external repository is reported in Table 4. The breakdown of these numbers by discipline is also available in Supplementary file 4 (A13D). This is an area of interest to the first author and will be the subject of future research. Some text has been added to contextualise these findings with respect to some of the studies raised in the reviewer’s comment.

5) The practice of editors editing reports when they disagree with the recommendation needs to be discussed further – even if it is only to say something along the lines that it is not clear if this involved deleting comments like "I support publication" or "I oppose publication", or if it was more extensive, and that unfortunately there were no open responses on this topic.

Another good point. Unfortunately, due to the design of the survey we are unable to clarify exactly what changes editors had in mind when they answered this question. It is plausible that the question may have been interpreted as changing small elements of the report if the editor disagreed with the recommendation, all the way up to switching a recommendation from “accept” to “reject”, or vice-versa. Consequently, making strong conclusions about likely behaviours from these results would be inappropriate. As advised, some text has been added to highlight possible differences in interpretation of the question. Furthermore, some text has been added to share two of the authors’ previous experience of uncovering an altered recommendation, as well as a link to a recent article in *Science* that covered the story in more detail. (Discussion paragraph six).

Also, please clarify what the figures (8-62%) refer to.

These figures are derived from Table 3. They communicate the findings that, depending on the situation, the proportion of editors that stated they would alter a reviewer's report without their permission in this study ranged from 8% (if they disagreed with the recommendation) to 62% (if comments for editor were left in the report). This sentence has been amended slightly to improve the clarity.

6) The authors could have perhaps included some more explicit recommendations for further analysis and even policy and practice.

The authors had initially considered including some recommendations for the broader scientific community, however felt this would be inappropriate given the spirit of the study. We see this study as an attempt to improve understanding of the state and perceptions of peer review in order to stimulate further discussion about peer review, catalyse future research in the area, and importantly, pave the way for policy pieces by researchers who specialise in policymaking and guideline development. Some text has been added into the Conclusion to better highlight this position.

7) Figure 1: Please replace Figure 1 with a table A6 from supplementary file 3.

This has been amended as requested (refer to Table 2 in the revised document).